# FRAGMENTFLOW: SCALABLE TRANSITION STATE GENERATION FOR LARGE MOLECULES

**Ron Shprints[1], Peter Holderrieth[1], Juno Nam[2],**
**Rafael Gómez-Bombarelli[2], Tommi Jaakkola[1]**
[1]MIT CSAIL, [2]MIT DMSE

## ABSTRACT

Transition states (TSs) are central to understanding and quantitatively predicting chemical reactivity and reaction mechanisms. Although traditional TS generation methods are computationally expensive, recent generative modeling approaches have enabled chemically meaningful TS prediction for relatively small molecules. However, these methods fail to generalize to practically relevant reaction substrates because of distribution shifts induced by increasing molecular sizes. Furthermore, TS geometries for larger molecules are not available at scale, making it infeasible to train generative models from scratch on such molecules. To address these challenges, we introduce FragmentFlow: a divide-and-conquer approach that trains a generative model to predict TS geometries for the *reactive core* atoms, which define the reaction mechanism. The full TS structure is then reconstructed by re-attaching substituent fragments to the predicted core. By operating on reactive cores, whose size and composition remain relatively invariant across molecular contexts, FragmentFlow mitigates distribution shifts in generative modeling. Evaluated on a new curated dataset of reactions involving reactants with up to 33 heavy atoms, FragmentFlow correctly identifies $90\%$ of TSs while requiring $30\%$ fewer saddle-point optimization steps than classical initialization schemes. These results point toward scalable TS generation for high-throughput reactivity studies. [Github], [Data].

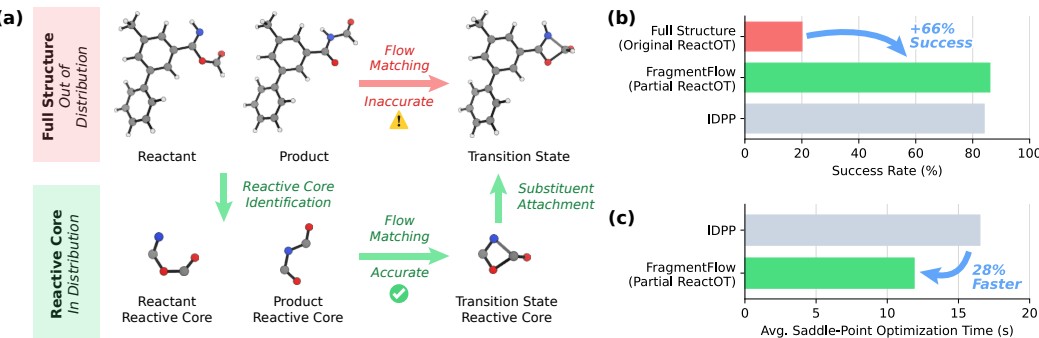

Figure 1: **FragmentFlow: A Fragmentation Based Approach for Transition State (TS) Generation**. **(a)** Given the reactants and products of chemical reactions, we use a flow matching model to generate the TS geometry of the reactive core only, which can be completed to the full structure upon the attachment of the substituents. While the generation of the full TS geometry is prone to errors that stem from a distribution shift for large molecules, FragmentFlow introduces an alternative that keeps the generative task in the training distribution. **(b)** FragmentFlow outperforms direct TS generation on full molecular structures, which suffers from a distribution shift. The success rate is defined as the fraction of molecules that are within 1 kcal mol$^{-1}$ of reference structures after saddle-point optimization (see Figure 4(a)). **(c)** FragmentFlow achieves a lower average wall-clock time required for Sella TS optimization per structure on LargeT1x (see Section 3.4), which is the most time consuming step in our TS generation procedure.

# 1 INTRODUCTION

Transition states (TSs) are critical structures for understanding chemical reactions, representing the highest-energy points along reaction pathways and corresponding to first-order saddle points on the potential energy surface (PES) (Vanden-Eijnden et al., 2010). Knowledge of the TS enables the study of the intrinsic energetic profile of a chemical reaction (Heid & Green, 2021; Karwounopoulos et al., 2025; van Gerwen et al., 2024; Spiekermann et al., 2022), which can be used to predict macroscopic observables such as reaction yield (Schwaller et al., 2021b), rate (Komp et al., 2022), and selectivity (Guan et al., 2021). Recently, TS structures have also been applied to the design of novel inorganic catalysts (Kwon et al., 2018; Maley et al., 2020) and enzymes (Ahern et al., 2025; Braun et al., 2025). Accurate and efficient identification of TS structures is therefore essential for high-throughput screening of chemical reactions and the design of new functional materials.

Optimization algorithms such as nudged elastic band (NEB) methods, string methods, and growing string methods can be used to systematically search for TSs on the PES (Weinan et al., 2002; Peters et al., 2004; Sheppard et al., 2008). However, these algorithms, coupled with electronic structure methods such as Density Functional Theory (DFT), are prohibitively expensive (Mardirossian & Head-Gordon, 2017). For example, for reasonably sized reaction networks, thousands of TS structures need to be optimized, requiring millions of single-point calculations, making classical approaches infeasible for high-throughput screening of chemical reactions (Van de Vijver & Zádor, 2020; von Lilienfeld et al., 2020; Margraf et al., 2023).

Machine learning (ML) methods have shown great promise in accelerating these computations (De Landsheere et al., 2026). In particular, it has recently been demonstrated that TS geometries can either be learned directly or optimized using generative models such as diffusion and flow matching (Duan et al., 2025; 2023; Darouich et al., 2025). A core limitation of this training paradigm lies in the training data. As shown in Table 1, the sizes of molecules represented in common datasets is rather small and significantly smaller than reactions of interest in applications such as drug discovery and sustainability. This poses a chicken-and-egg problem: generative models require TS geometries as training data, yet these structures are prohibitively expensive to calculate for large molecules.

This work introduces **FragmentFlow**, which tackles the aforementioned distribution shift problem by applying a divide-and-conquer strategy (see Figure 1). First, given the geometries of the reactants and products, we identify the reactive core (i.e., a fragment of the reactive molecules that is modified during the reaction). Second, we use a flow matching model to generate a candidate TS geometry for the reactive core only, which matches the distribution of molecular sizes in the training data. Finally, after the TS geometry for the reactive core is generated, we re-attach the substituent groups that do not participate in the reaction to recover the full molecular structure.

Our *core hypothesis* is that the reactive core exerts the greatest influence on the geometry of the transition state. Consequently, it must be modeled with high accuracy, whereas the substituents need not be treated with the same level of computational effort. This stratified modeling strategy is inspired by approaches in other areas of computational chemistry. For example, computational enzymology leverages QM/MM simulations where the reactive species are treated using quantum mechanical methods while the surrounding enzyme environment is modeled with simpler molecular mechanical methods (Van Der Kamp & Mulholland, 2013). Moreover, analogous ideas appear in protein generation, where backbone atoms are often generated first and side chains are subsequently completed (Geffner et al., 2025; Ingraham et al., 2023; Jumper et al., 2021; Yim et al., 2023).

To demonstrate that FragmentFlow facilitates the generation of TS geometries for large molecules, we curate **LargeT1x**: a new evaluation dataset comprising 131 reactions involving molecules with up to 33 heavy atoms (see Section 3.4). On this dataset, FragmentFlow produces approximately $90\%$ of TS structures within $1$ kcal mol$^{-1}$ of the reference structures after Sella saddle-point optimization (Hermes et al., 2022). Moreover, this accuracy is achieved with $30\%$ fewer Sella optimization steps compared to initial guesses from classical methods such as IDPP (Schmerwitz et al., 2023). This reduction translates into up to a $28\%$ improvement in wall-clock time on a 128-CPU machine, bringing TS generation closer to the time requirements of high-throughput screening applications.

In summary, our main contributions are as follows:

1. **A new paradigm for scalable TS generation.** We introduce FragmentFlow, a generative modeling framework for TS prediction that is specifically designed to handle reactions involving large molecules and mitigate size-induced distribution shifts.

2. **A new benchmark and empirical validation.** We present a curated dataset of reactions with reactants, products, and TS geometries containing up to 33 heavy atoms, substantially extending existing TS benchmarks. Using this dataset, we demonstrate that FragmentFlow significantly improves TS generation success rates and achieves high quality TSs with better time efficiency than existing methods.

3. **Validation of the reactive core scaling hypothesis.** We confirm a strong correlation between the quality of the generated reactive core and the accuracy of the full TS geometry.

## 2 RELATED WORKS

**Flow matching (FM)** is a generative modeling framework that transforms samples from a simple base distribution $p_0$ into a complex target distribution $p_1$, which approximates an underlying data distribution $q$ from which only a limited number of samples are available. This transformation is achieved via a time-dependent diffeomorphic map, known as a flow, $\psi : [0, 1] \times \mathcal{X} \to \mathcal{X}$. The flow is defined by the following differential equation: $\partial_t \psi_t(x) = u_t(\psi_t(x))$ with $\psi_0(x) = x$, where $u_t : [0, 1] \times \mathcal{X} \to \mathcal{X}$ is a vector field governs the evolution of the $\psi_t$. The flow induces a time-dependent probability density path $p_t : [0, 1] \times \mathcal{X} \to \mathbb{R}$ starting at $p_0$ and ending with $p_1$. FM trains a parametric approximation $u_t^\theta$ of the true vector field $u_t$ by solving a regression objective (Lipman et al., 2024; 2022). For TS generation, FM models are trained on triplets of reactants, products, and TS geometries computed using methods such as NEB, thereby learning a direct mapping from reactants and products to the corresponding TS and significantly accelerating the generation process. In the double-ended setting, as considered in this work, the input to these models is often a linear interpolation between the reactants and products (Duan et al., 2025).

**TS generation** using generative models has been explored along several directions. Mitchell & Rotskoff (2024) adopt a Deep Q-Learning approach, where the Committor Function is learned over the state space (Mnih et al., 2015). This allows for the sampling of TSs that are characterized by a Committor value of 0.5. An alternative approach frames the problem as that of transition path sampling by learning Doob's $h$-transform through a variational optimization over trajectory distributions (Du et al., 2024). Lastly, Duan et al. (2023) introduce OA-ReactDiff, which uses a diffusion process to learn a mapping between an initial guess and true TSs. This work is further improved by the ReactOT model from Duan et al. (2025), which removes the stochasiticty from OA-ReactDiff and enables faster generation of TSs. Several other works use either diffusion or FM models for TS generation, but as the efficiency of the generation process is critical for possible applications, we choose ReactOT as the main backbone for our modeling approach (Kim et al., 2024; Zhao et al., 2025).

**Distribution shifts** in the context of TS generation have recently been investigated by Darouich et al. (2026), who focus on distribution shifts arising from variations in molecular composition, such as atom types. They address this challenge by introducing a self-supervised pretraining strategy for generative TS prediction that leverages conformers of equilibrium structures. Here, we adopt a complementary approach and focus on distribution shifts induced by variations in molecular size. Beyond demonstrating the effectiveness of our approach in terms of both accuracy and efficiency, we provide empirical scaling laws with respect to molecular size. Our modeling strategy is most closely related to fragment or motif based representations used in modeling small molecules (Poletukhin et al., 2026; Jin et al., 2018). To the best of our knowledge, this work is the first to apply such an approach to the TS generation problem.

## 3 METHODS

### 3.1 OUT OF DISTRIBUTION GENERALIZATION

We begin by explaining the core theoretical motivation of this work. Let $x \in \mathbb{R}^{N \times 3}$ be a TS geometry with $N$ atoms. Our goal is to model a distribution $q(x|y_R, y_P)$ over TSs given reactants and products $y_R, y_P$. For better readability, we drop the conditioning from $q(x|y_R, y_P)$ and simply write it as a distribution $q(x)$ over TSs since the following argument will not depend on it.

A single global model $p_{\text{full}}(x)$ must fit all coordinates, and the full modeling error $D_{\text{KL}}(q(x)\|p_{\text{full}}(x))$ increases with $N$ as more degrees of freedom contribute to it (see Appendix B). In contrast, our approach exploits the *locality* of reactive sites by splitting $x = (x_c, x_s)$ into a small reactive core $x_c$ and substituents $x_s$, and parameterizing a factorized model $p_{\text{fact}}(x) = p(x_c)\,p(x \mid x_c)$. For this factorized model, the chain rule for the KL-divergence gives

$$\underbrace{D_{\text{KL}}\big(q(x)\,\|\,p_{\text{fact}}(x)\big)}_{\text{Full modeling error}} = \underbrace{D_{\text{KL}}\big(q(x_c)\,\|\,p(x_c)\big)}_{\text{(1) Reactive core modeling error}} + \underbrace{\mathbb{E}_{q(x_c)}\big[D_{\text{KL}}\big(q(x \mid x_c)\,\|\,p(x \mid x_c)\big)\big]}_{\text{(2) Attachment error}}.$$

Here, the (1) reactive core modeling error is governed primarily by the size of the reactive core, $N_c \ll N$. Meanwhile, the (2) attachment error does not necessarily increase with the number of additional substituents when their coordinates are approximately inherited from the reactant and product structures and only a small interfacial region is predicted. In particular, we hypothesize that the attachment error grows slowly with molecular size, as it relies on conventional computational chemistry approximations rather than data-driven learning. Consequently, in contrast to global TS generation, the effective modeling overhead is determined by the localized core and the boundary complexity rather than the full molecular structure, enabling improved out-of-distribution generalization to larger values of $N$.

## 3.2 GENERATION OF TS GEOMETRIES FOR REACTIVE CORE FRAGMENTS

ReactOT (Duan et al., 2025) is a FM model that generates transition state structures by formulating TS prediction as an optimal transport problem. Given a reaction with reactants $R$ and products $P$, ReactOT learns to transport samples from an initial distribution $p_0$ to a target distribution $p_1$, which approximates the underlying distribution $q$ of TS structures. The key insight is to leverage the pairing information between initial guesses and ground-truth TSs by setting $x_0 = (R + P)/2 \sim p_0$ and $x_1 = \text{TS} \sim q$. This choice of initial guess is motivated by the assumption that the midpoint between the reactants and products provides a reasonable starting point close to the TS geometry. Under the optimal transport framework, the optimal trajectories between paired samples $(x_0, x_1) \sim \pi^*$ (where $\pi^*$ is the optimal transport coupling) move along straight lines $x_t = (1 - t)x_0 + tx_1$, yielding a constant velocity vector field $u_t(x_t) = x_1 - x_0$. ReactOT trains a neural network $u_t^\theta$ to approximate this vector field by minimizing the flow matching objective:

$$\min_\theta \mathbb{E}_{(x_0, x_1) \sim \pi^*} \left[\|u_t^\theta(x_t, t, z) - (x_1 - x_0)\|^2\right],$$

where $z$ represents conditional information such as the atom types. At inference, the TS structure is generated by solving the ordinary differential equation $\partial_t x_t = u_t^\theta(x_t, t, z)$ starting from $x_0$.

**Partial ReactOT.** To enable our model to generate TS geometries of reactive cores only, we train the ReactOT model on the Transition1x training set that we augment with fragmented structures created according to the procedures in Sections 3.4 and 4.1 (see Appendix A for additional training hyperparameters). To model fragmented reactive cores we mask the substituent atoms and minimize the following loss function:

$$\min_\theta \mathbb{E}_{(x_{c,0}, x_{c,1}) \sim \pi_c^*} \left[\|u_t^\theta(x_{c,t}, t, z_c) - (x_{c,1} - x_{c,0})\|^2\right],$$

where $x_{c,1} \sim q(x_c)$ is the masked geometry, $x_{c,0}$ is the masked average interpolation of the reactants and products, and $z_c$ is the masked conditioning. The utility of this approach is illustrated in Section 4.1 and Appendix D, where we show that this data augmentation significantly improves the performance of FragmentFlow. We hypothesize that introducing this data augmentation method allows the model to learn how to generate TS structures with missing connectivities. This in turn allows for better generation of TS geometries for reactive cores while keeping the generation process within the training distribution. We refer to the resulting model as **Partial ReactOT**.

## 3.3 SUBSTITUENT ATTACHMENT AND REFINEMENT

We discuss how substituents are re-attached to the generated reactive core structure to obtain the full TS geometry. After generating the reactive core of a TS, we use IDPP interpolation to construct 10

candidate full-molecule structures. From these candidates, we select the structure whose reactive core minimizes the RMSD to the generated reactive core from Partial ReactOT. We then apply Kabsch alignment to align the reactive cores and replace the reactive core in the selected IDPP structure with the generated reactive core. The motivation for this procedure is that, while the generated reactive core more accurately represents the TS geometry, the IDPP-generated structure is obtained through a globally coherent interpolation, resulting in physically plausible substituent placements. Although this coherence may be partially disrupted by replacing the reactive core, we use this step as an approximation.

Importantly, the subsequent refinement step is common to *all* evaluated methods including IDPP and vanilla ReactOT: in all cases, the resulting structure is further optimized using the Sella TS optimizer. This step improves the TS geometry and mitigates inconsistencies introduced during the generation process.

### 3.4 DATASET CONSTRUCTION

In order to evaluate our method on reactions involving large molecules, we require a benchmark dataset that contains TS geometries at large molecular scales. However, existing TS datasets are largely restricted to small molecules (see Table 1). To address this gap, we construct **LargeT1x**: a reaction dataset comprising TS geometries for molecules larger than those typically found in existing datasets. Our workflow, summarized in Figure 2, begins with geometries from the Transition1x validation set and applies processing and validation steps to obtain the final geometries (Schreiner et al., 2022).[1]

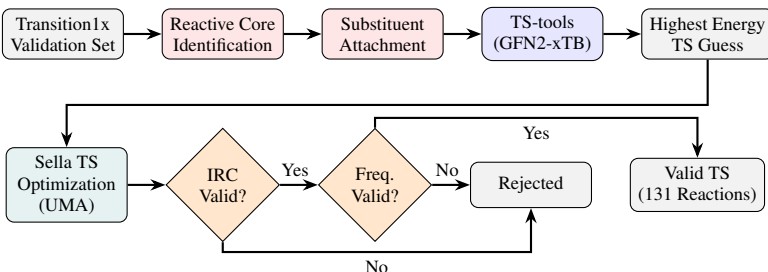

Figure 2: **Dataset Curation Workflow.** The different stages of the dataset curation consist of: reactive core identification, TS generation with TS-tools, Sella optimization using UMA, and two-stage validation (IRC + vibrational frequency analysis).

**Reactive Core Identification.** We first converted the 3D geometries from Transition1x to SMILES representations using the xyz2mol package (Kim & Kim, 2015). For each reaction, we identified the reactive core using the Bemis–Murcko scaffold identification to detect the backbone structure (Bemis & Murcko, 1996). Subsequently, we use the Weisfeiler-Lehman Network (WLN) atom mapper to distinguish reactive atoms from unreactive substituents (Jin et al., 2017). The final reactive core consists of the union of the backbone atoms identified in the first step and the reactive atoms identified in the second step. We then systematically detached substituent groups from the reactive core and re-attached larger inert substituents that we have curated (see Appendix F).

**TS Generation.** We used the TS-tools package to generate initial TS guesses using the GFN2-xTB method together with reactant and product geometries (Stuyver, 2024; Bannwarth et al., 2019). TS-tools constructs reactive complexes by aligning reactants based on atom-mapped reaction SMILES, where forming bonds are pre-stretched to facilitate the reactant to product transformation. Artificial force-induced reaction paths are then generated by applying harmonic potentials to the forming bonds, with force constants incrementally adjusted until the system traverses from the reactant to the product basin. Up to 5 preliminary guesses are retained from local maxima along these biased paths, ranked by their energies and filtered by imaginary frequency magnitude and mode alignment with active bonds. The structure showing the highest energy along the reaction path is considered the most promising TS candidate. Therefore, we choose this structure and refine it further using the following optimization procedure.

---

[1]We use the same training and validation split of Transition1x as was used by Duan et al. (2025).

**TS Optimization and Validation.** The highest energy preliminary TS structure is refined using the Sella TS optimizer with UMA as the energy evaluator (we use the same optimization hyperparameters as in Section 4).[2] This combination enables more efficient optimization while maintaining accuracy comparable to classical methods. Following optimization, each TS structure is subjected to a two-stage validation procedure.

First, we perform intrinsic reaction coordinate (IRC) calculations to verify connectivity to the correct reactant and product states. We compute the RMSD between each IRC endpoint and the corresponding reactant and product geometries. A reaction is retained only if one IRC endpoint shows lower RMSD to the reactants while the other shows lower RMSD to the products, with an RMSD difference threshold of 0.3 Å to ensure clear endpoint assignment.

Second, we perform vibrational frequency analysis using UMA as the calculator. We verified that each TS candidate has exactly one significant imaginary frequency (below $-50\,\mathrm{cm}^{-1}$). Additionally, we analyze the displacement vector of the imaginary mode to confirm that it corresponded to the active bonds identified during reaction preparation. Specifically, we displace the TS geometry along the imaginary mode, compute pairwise distance changes, and verify that the bond exhibiting maximum displacement was among the bonds that are either formed or broken during the reaction. Only structures that meet both of the IRC consistency and vibrational analysis criteria are retained as valid TSs.

**Post-processing.** ReactOT achieves SE(3) equivariance through both model architecture and data pre-processing. The model architecture is based on LEFTNet (Du et al., 2023), an SE(3)-equivariant graph neural network (GNN). LEFTNet is enhanced with object-aware improvements to handle chemical reactions, distinguishing between reactant and product molecules while maintaining equivariance to rotation, translation, and permutation transformations (Duan et al., 2023). To ensure equivariance during training, the data is pre-processed by aligning reactants and products using the Kabsch algorithm, removing arbitrary orientations between reactants and products. Additionally, the center of mass of each molecular system is shifted to the origin to remove translational degrees of freedom. We use the same post-processing procedure on LargeT1x to ensure SE(3) equivariance.

**Dataset Statistics.** Of 2200 TSs that were originally optimized with Sella, 131 reactions passed the validation step. Those reactions correspond to 33 unique reactions from the Transition1x validation set. While several reactions failed during the TS generation and optimization steps, the vast majority of them were lost during the validation steps, as we further discuss in Appendix C. While every data point is valuable in this data-scarce setting, we favor strict validation criteria to ensure the rigor of our conclusions. Importantly, as the dataset serves only as an evaluation benchmark, a relatively small number of reactions is adequate. The range of heavy atoms in the reactive molecules is given in Table 1, where it is compared to other common publicly available datasets. We use LargeT1x for our evaluations in Section 4.

Table 1: **Dataset Comparison.** The range of heavy atoms in common datasets that contain reactants, products, and TS geometries. Datasets with relatively small molecules are abundant, but such datasets get increasingly challenging to generate at scale as the molecule size gets larger. The relatively small number of unique reactions in LargeT1x is explained by stringent validation thresholds and rigorous filtering steps as described in section 3.4.

| Dataset | Range of Heavy Atoms | Number of Unique Reactions |
|---|---|---|
| Lee et al. (2025b) | 7-8 | 19,000 |
| Lee et al. (2025a) | 2-10 | 196,979 |
| Schreiner et al. (2022) | 3-7 | 10,073 |
| LargeT1x (ours) | 10-33 | 131 |

## 4 EXPERIMENTS

In this section, we evaluate the *core hypothesis* introduced in Section 3.1. We begin by outlining the full experimental setting of FragmentFlow. Then, we compare several TS generation methods

---

[2]We benchmark UMA in this problem setting in Appendix E.

and show that, among the models we tested that satisfy our quality criteria, FragmentFlow is the most efficient. Finally, we analyze these results and demonstrate that the superior performance of FragmentFlow arises from its ability to generate higher quality geometries for the reactive core, thereby confirming the *core hypothesis*.

In our experimental setup, we sample TS geometries from ReactOT using a fixed number of function evaluations (NFEs) of 200, and we use a force convergence criterion of 0.05 with a maximum of 500 optimization steps for the Sella optimizer.[3] Although the sampling time increases with the number of NFEs, it remains negligible compared to the cost of Sella optimization–particularly in our setting, where only relatively small reactive cores are sampled faster than full molecular structures, which require more time. Consequently, we use the number of Sella optimization steps as the primary evaluation metric throughout this section. At inference time, TS optimization constitutes the main computational bottleneck, often requiring several minutes per structure, which is several orders of magnitude longer than ReactOT sampling. Therefore, our objective is to **construct improved initial guesses that substantially reduce the number of Sella optimization steps** required.[4]

## 4.1 MODEL ABLATIONS

We train our model on the training set of Transition1x that is augmented by fragmented reactive cores. In addition to that, we introduce random fragments, which include at least 60% of the atoms in the molecules.[5] The motivation behind this step is to further expose the model to structures with missing connectivities, while increasing the likelihood that the true reactive core is covered when the automatic core extraction fails. The utility of this data augmentation approach is illustrated in Figure 3, which shows that the average number of Sella optimization steps on the full structures generated by FragmentFlow decreases with more random fragments in the training set. We attribute this result to the improved ability of the model to generate reactive cores accurately, as we further analyze in Section 4.3 and Appendix D. Following the ablation studies, we determine our experimental setting: we choose the model that is trained on 1000 epochs on the dataset with 6 additional random fragments per reaction. The full training recipe is provided in Appendix A.

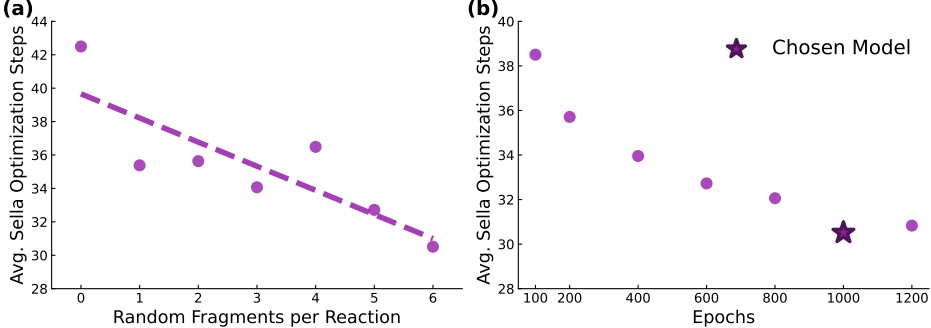

Figure 3: **Ablation Studies for Partial ReactOT: Examining the Average Number of Sella Optimization Steps.** **(a)** Adding additional random fragments on top of the reactive core improves the quality of our TS guesses. The tested models were trained until convergence. **(b)** For the dataset with six additional random fragments, the quality of the generated TS guesses improves with the number of epochs.

## 4.2 EVALUATIONS

We compare the quality and efficiency of different TS generation methods on LargeT1x, focusing on both ML and classical approaches. For ML methods, we evaluate ReactOT for generating full TS structures. As described in Section 3.1, it is hypothesized that ReactOT suffers from significant errors due to distribution shifts when applied to molecules larger than those in its training set. To test FragmentFlow, we use both Partial ReactOT (trained on our fragmented dataset) and the original

---

[3]In this case, NFEs are equivalent to the number of neural network evaluations of the flow model.

[4]We discuss the importance of the Sella optimization step in Appendix B.

[5]This value was determined based on the average proportion of kept atoms in the identified reactive cores in the validation set of Transition1x.

ReactOT checkpoint. For classical methods, we benchmark linear interpolation and IDPP. For these methods, we generate interpolations with 10 intermediate structures and use UMA to select the structure with the highest energy as the TS guess. All TS guesses are then refined using Sella optimization to yield the final TS structure.

Figure 4(a) shows the RMSD and absolute energy distributions obtained for each method. Notably, only FragmentFlow using Partial ReactOT and IDPP produce approximately $90\%$ of structures with energies within $1 \text{ kcal mol}^{-1}$ of the reference TS structures. This confirms that the original ReactOT checkpoint suffers from significant errors due to distribution shifts when applied to larger molecules. Furthermore, this result confirms that training on fragmented structures enables Partial ReactOT to predict TS geometries of reactive cores more accurately.

Next, we compare the efficiency of the best-performing methods. As shown in Figure 4(b), FragmentFlow requires $30\%$ fewer Sella optimization steps on average than IDPP. Figure 4(c) illustrates how these methods scale with molecular size, measured by the number of heavy atoms. Critically, the efficiency gap widens as the molecules grow larger, indicating that FragmentFlow is particularly advantageous for large molecules.

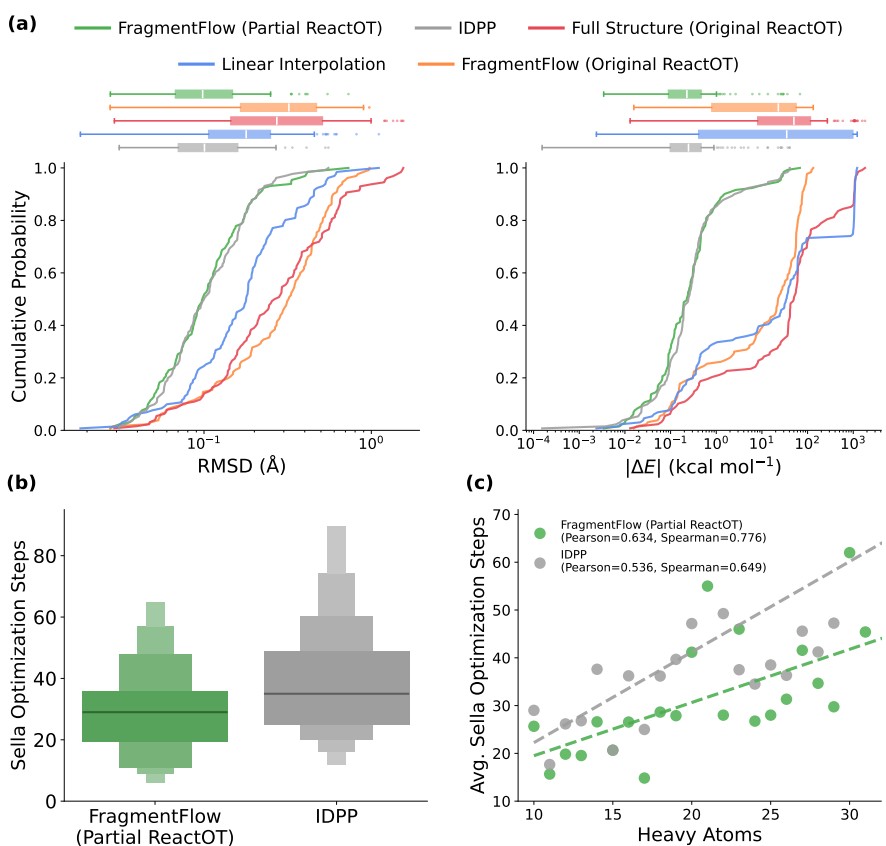

Figure 4: **TS Generation Quality and Efficiency.** **(a)** Empirical distributions of generated structures by RMSD and absolute energy difference to the reference TS structures. **(b)** FragmentFlow is more efficient than the IDPP method, despite having a comparable quality of structures. **(c)** The scaling laws of the average number of Sella optimization steps as a function of the number of heavy atoms in the molecules suggest that the FragmentFlow gets increasingly more efficient for larger molecules.

### 4.3 ANALYSIS

In this section, we test the *core hypothesis* and show that the reactive core has the greatest influence on TS geometry, which motivates the fragmentation-based modeling approach. To test this hypothesis, we examine the reactive cores from the tested TS generation methods prior to Sella optimization and show that their quality correlates with the number of Sella optimization steps required.

Figure 5(a) shows that FragmentFlow using Partial ReactOT produces reactive cores with the lowest RMSD to the reference TSs on average, among all the tested methods. To examine the relationship between reactive core RMSD and optimization efficiency, we plot these values for each structure in Figure 5(b). This figure demonstrates that FragmentFlow using Partial ReactOT produces a greater number of reactive cores with low RMSD to the reference TSs compared to IDPP. Critically, structures with more accurate reactive cores require substantially fewer Sella optimization steps, confirming that the quality of the reactive core correlates with the optimization efficiency.

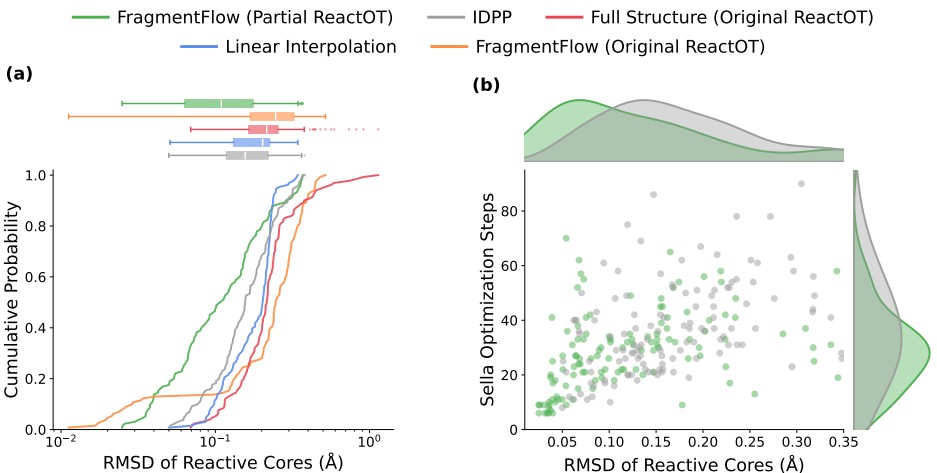

Figure 5: **Examining the *Core Hypothesis*.** **(a)** The empirical distribution of the RMSD between the generated reactive cores and the reference reactive cores. **(b)** Structures with reactive cores that are closer to the reference require less Sella optimization steps, which boosts the efficiency of FragmentFlow. FragmentFlow (Partial ReactOT) achieves a Pearson correlation of 0.3 and a Spearman correlation of 0.5. IDPP achieves a Pearson correlation of 0.14 and a Pearson correlation of 0.43.

## 5 Conclusions

We introduce FragmentFlow: a fragmentation-based approach for generating TS structures of large molecules, directly addressing the distribution shift problem that limits existing ML methods. Our *core hypothesis*, that the reactive core exerts the dominant influence on the TS geometry, was validated empirically. By training a generative model on fragmented molecular structures and re-attaching substituents after the generation of the TS reactive core, we demonstrate that FragmentFlow produces high-quality TS structures. Moreover, this approach requires $30\%$ fewer Sella optimization steps on average than the classical IDPP method. The correlation between reactive core accuracy and optimization efficiency validates our divide-and-conquer strategy and establishes a path forward for scalable TS generation.

We identify the following directions for future work. First, extending our approach to generate TS ensembles rather than single structures could improve its robustness. While the ReactOT model is deterministic due to the fixed initial condition $x_0 = (P + R)/2$, incorporating stochasticity into the generative model while keeping it as efficient as possible could be a major improvement. Second, the Partial ReactOT model could be further improved through architectural innovations or alternative training strategies tailored to molecular fragments to improve the quality of the generated reactive cores. Third, more sophisticated attachment methods beyond IDPP interpolation, such as a learned re-attachment method, could further reduce the required Sella optimization steps. This task can possibly be formulated as a graph inpainting task, similarly to image inpainting in computer vision (Xu et al., 2023; Lugmayr et al., 2022). In addition, expanding our dataset and training foundation models on diverse reaction types would be necessary to benchmark our approach on additional chemical spaces. Finally, the fragmentation-based modeling approach can be applied beyond the double-ended setting for additional tasks such as reaction prediction.

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

## ACKNOWLEDGMENTS

We would like to thank Thijs Stuyver for helpful discussions regarding the dataset curation.

# A   PARTIAL REACTOT TRAINING RECIPE

We start by reproducing the results for the original ReactOT model. Our model was trained on 8 A100 GPUs with dynamic batch sampling based on molecular size. We resume the training from the OA-ReactDiff (Duan et al., 2023) for 200 epochs as specified by the training recipe in (Duan et al., 2025). We compare our results to those we obtain from the provided ReactOT checkpoint in Figure 6 and conclude that we can replicate the reported results in (Duan et al., 2025) to a good extent.

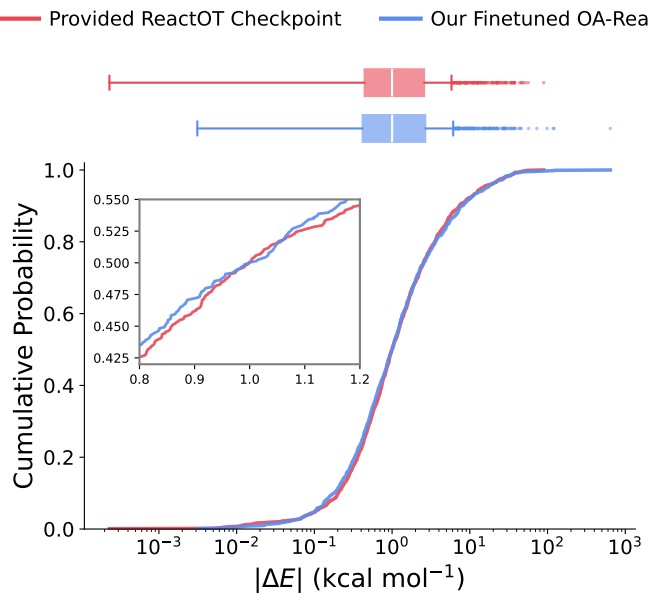

Figure 6: **Comparing Our Trained Checkpoint with the Provided Checkpoint for ReactOT.** $|\Delta E|$ evaluations on the validation set of Transition1x using the same procedure and training settings as used in (Duan et al., 2025) (NFE = 50). We can recover nearly identical results by resuming training from the OA-ReactDiff checkpoint.

Next, we provide the training recipe that we use to train Partial ReactOT. The model is trained on the Transition1x training set augmented with fragmented structures as described in Section 4.1. For training Partial ReactOT, we use a maximum of 10,000 atoms per batch. We use the same data pre-processing steps: we apply SE(3) equivariance by aligning reactants and products using the Kabsch algorithm and shifting the center of mass to the origin. Table 2 summarizes the key hyperparameters and training configuration.

Table 2: **Training Recipe for Partial ReactOT.** Summary of model architecture, diffusion configuration, optimization, and training hyperparameters used for training.

| Hyperparameter | Value |
|---|---|
| *Model Architecture* | |
| Backbone | LEFTNet |
| Number of parameters | 10,645,719 |
| Initial checkpoint | OA-ReactDiff |
| *Diffusion Configuration* | |
| Timesteps | 3000 |
| Mapping | $\frac{R+P}{2} \rightarrow$ TS |
| Sampling method | OT-ODE |
| *Optimization* | |
| Optimizer | AdamW |
| Learning rate | $1 \times 10^{-5}$ |
| Weight decay | 0 |
| Betas | (0.9, 0.999) |
| Batch size (max nodes) | 10,000 |
| *EMA* | |
| EMA decay | 0.999 |
| EMA start step | 2000 |
| *Training Schedule* | |
| Learning rate schedule | Constant |
| Warmup epochs | None |
| Loss function | L2 (MSE) |

# B  EXAMINING THE PERFORMANCE OF REACTOT

We further motivate our method with supplementary results on the original ReactOT model. ReactOT generates TS structures in approximately 0.4 seconds, achieving an accuracy within 1 kcal mol$^{-1}$ of the reference structures in approximately $50\%$ of cases for NFE = 50. This success rate increases to $81\%$ after Sella optimization (see Figure 7). However, Sella optimization can require several minutes per structure, increasing the overall TS generation time by orders of magnitude. Moreover, this performance gap cannot be closed by increasing the number of NFEs alone (see Figure 8). These results suggest that Sella optimization should be viewed as a central component of the TS generation workflow.

Furthermore, we show that the performance of ReactOT degrades on larger molecules in the validation set of Transition1x. As demonstrated by Figure 9, larger molecules in this set incur larger $|\Delta E|$ values when compared to the reference TS structures. Interestingly, this happens in contrary to the dominance of molecules with the same size in the training set (see Figure 11). We hypothesize that this pattern arises from small structural deviations in the generated TSs, which occur more frequently for larger molecules and can translate into significant energy gaps, as even minor geometric perturbations can have a substantial impact on TS energetics. This trend is also observed when the number of the required Sella optimization steps is considered, which is higher for larger molecules in this dataset (see Figure 10). Overall, this failure mode of ReactOT motivates a special treatment for TS generation of large molecules, as we propose in this work.

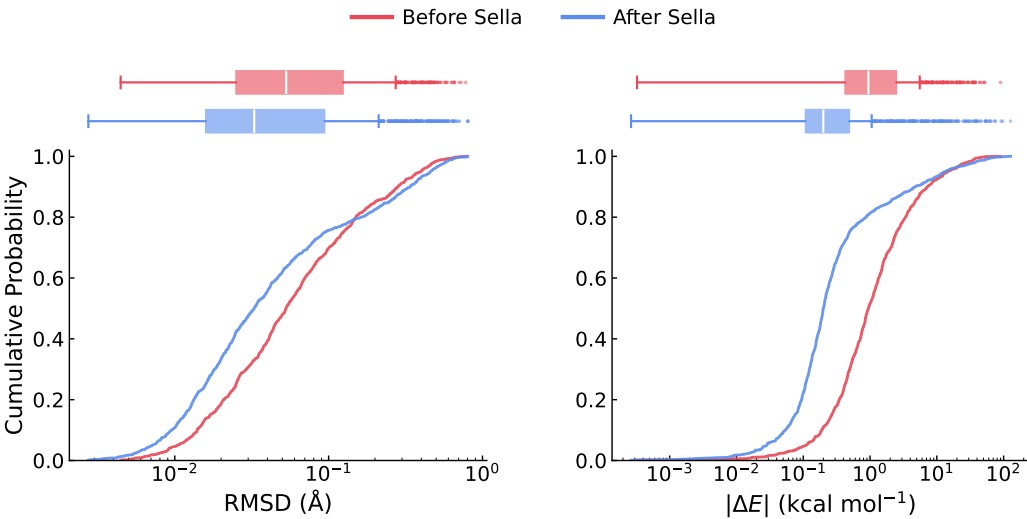

Figure 7: **The Effect of Sella Optimization on the Transition1x Validation Set.** Optimizing the TSs from ReactOT (NFE = 200) can significantly improve the quality of the generated structures.

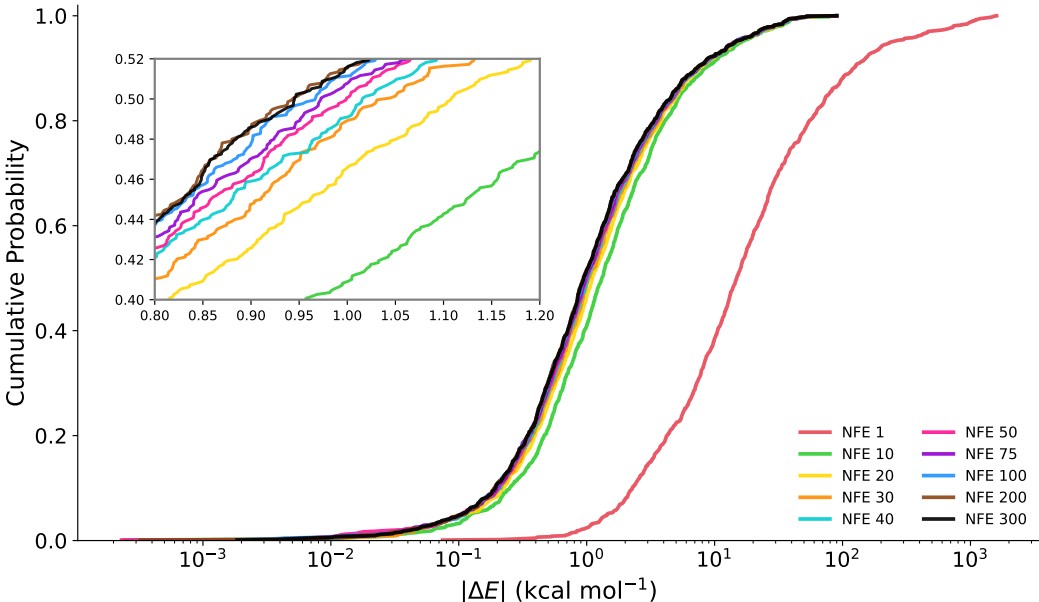

Figure 8: **The Effect of the Number of NFEs on the Performance of ReactOT on the Validation Set of Transition1x.** Taking more NFEs improves the quality of the generated TSs. However, the performance gain plateaus after 200 NFEs. These results are prior to Sella optimization.

# C  ADDITIONAL DATASET STATISTICS

We provide additional statistics on LargeT1x. Following the reactive core identification step, 483 reactive cores were successfully identified out of the 1073 reactions in the Transition1x validation set. The remaining reactions didn't pass our confidence filters, which ensured mild sanity checks (e.g., an atom in the reactants has to be mapped to the same atom type in the products). While the original validation set contains molecules with 10-69 atoms and a mean of 39.7 atoms, the reactive cores contain merely 5-7 atoms with a mean of 6.3 atoms (including hydrogen atoms).

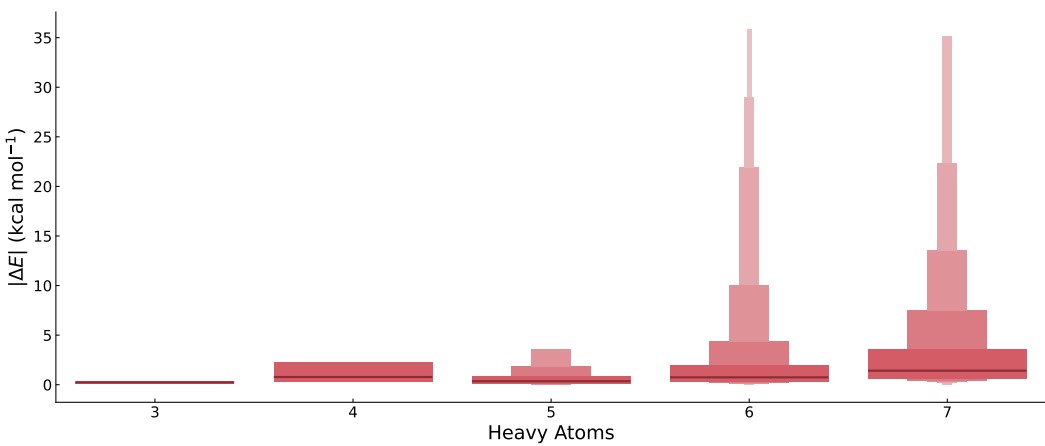

Figure 9: $|\Delta E|$ **Distributions on the Validation Set of Transition1x (Before Sella Optimization) per TS Size.** ReactOT generates mostly highly accurate TSs for small molecules in this set (NFE $=200$), but is prone to errors as the size of the molecules gets larger, even when the size is in the training distribution.

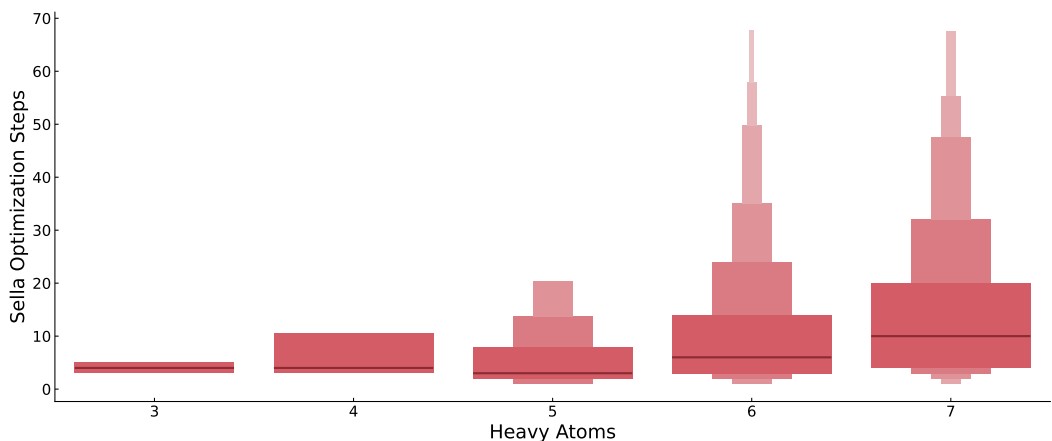

Figure 10: **Number of Sella Optimization Steps Required for Generated TSs from the Validation Set of Transition1x per TS Size.** Larger TS structures from ReactOT (NFE $=200$) require more Sella optimization steps on average than smaller structures. This highlights the difficulty of generating accurate TS structures for large molecules.

For the atom mapper, we tested two models: WLN mapper and RXNMapper (Schwaller et al., 2021a). After inspecting the mapping results on a few dozen randomly chosen reactions from the validation set of Transition1x, we concluded that WLN mapper achieves better performance, while RXNMapper yields wrong mappings in most of the tested reactions (see Figure 21 for a few examples). Therefore, we chose to use WLN mapper in this work. To further improve the robustness of our reactive core identification method, we also take the union with the backbone atoms as identified by the Bemis–Murcko scaffold (Bemis & Murcko, 1996). Figure 22 illustrates the effect of this step.

Most of the reactions were filtered during the IRC validation step, reducing the dataset size roughly to its final size. The reactions that were filtered out include reactions with inconsistent IRC endpoints and those with IRC endpoints with an RMSD higher than $0.3$ Å to either the reference reactants or products geometries. After filtering, the remaining reactions have an average RMSD to the IRC endpoints of $0.15$ Å with a median of $0.14$ Å and standard deviation of $0.06$ Å.

In the final dataset, 60 of the reactions result from mono-substitution, 43 from di-substitutions, and 28 from tri-substitutions. The substitution type is chosen randomly based on the number of identified substitution locations on the reactive core. For mono-substituted reactions, we bias the sampling of substituents based on the number of the atoms in the substituents (with heavier substituents being

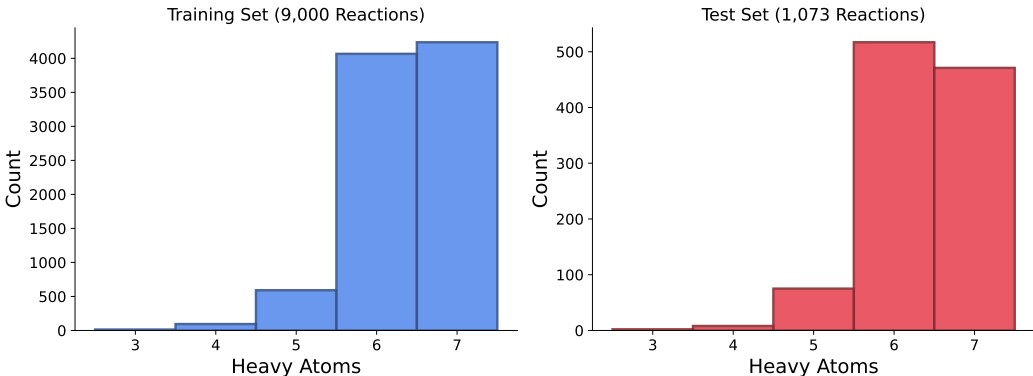

Figure 11: **The Distribution of Heavy Atoms in the Transition1x Training and Validation Sets.** Although the distributions seem to match, ReactOT still incurs more significant errors on larger molecules than on smaller ones. This is in contrary to the fraction of such molecules in the training set.

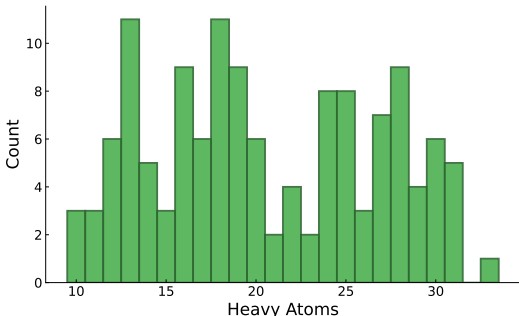

Figure 12: **Distribution of Heavy Atoms in LargeT1x.** This evaluation dataset is dominated by molecules that are larger by several factors than those in Transition1x.

more likely). Otherwise, we sample substituents at random (see Figure 23 for the structures of the set of substituents that we use for the attachment step). Each reactive core is substituted up to 5 times to allow for diverse substitution patterns, while ensuring that each substitution is unique (per reactive core). The distribution of molecular sizes in LargeT1x is presented in Figure 12.

Finally, we compare the performance of ReactOT and IDPP on the original 33 reactions from the validation set of Transition1x, from which the final LargeT1x dataset was derived. The results (before Sella optimization) are presented in Figure 13. This illustrates that these reactions are not trivially modeled by the IDPP baseline, highlighting the efficacy of FragmentFlow in modeling their substituted counterparts.

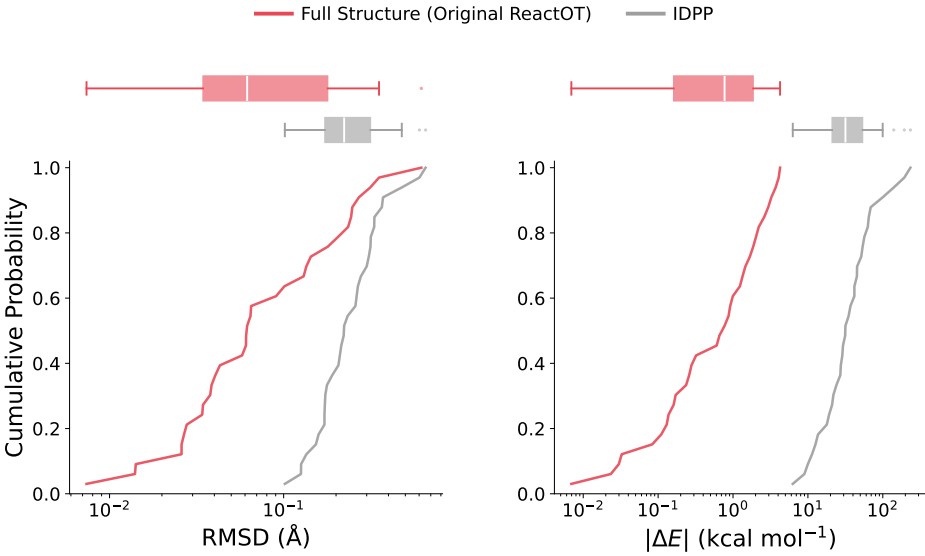

Figure 13: **Testing ReactOT on the Original 33 Reactions from the Transition1x Validation Set.** Those reactions are not trivially modeled by IDPP and ReactOT achieves must better TS structures. For comparison IDPP yields no structures within 1 kcal mol$^{-1}$ of the reference TSs.

## D    FULL EVALUATIONS

For completeness, we present the full evaluation results as a complement to Figure 4. The distributions of the number of Sella optimization steps are presented in Figure 14 and the statistics are presented in Table 3. In addition, we note that optimizing the generated structures from Fragment-Flow (Partial ReactOT), IDPP, and Full Structure (Original ReactOT) took 26 minutes, 36 minutes, and 3 hours on the same 128-CPU machine, respectively. Lastly, we present the scaling laws with respect to number of heavy atoms for all the methods in Figure 15. Notably, when comparing the results of Full Structure (Original ReactOT) and Linear Interpolation, we notice that the latter scales better with size despite being the simplest method, emphasizing the significance of the distribution shift. These results highlight the efficacy of FragmentFlow (Partial ReactOT), which is the fastest and most accurate TS generation method.

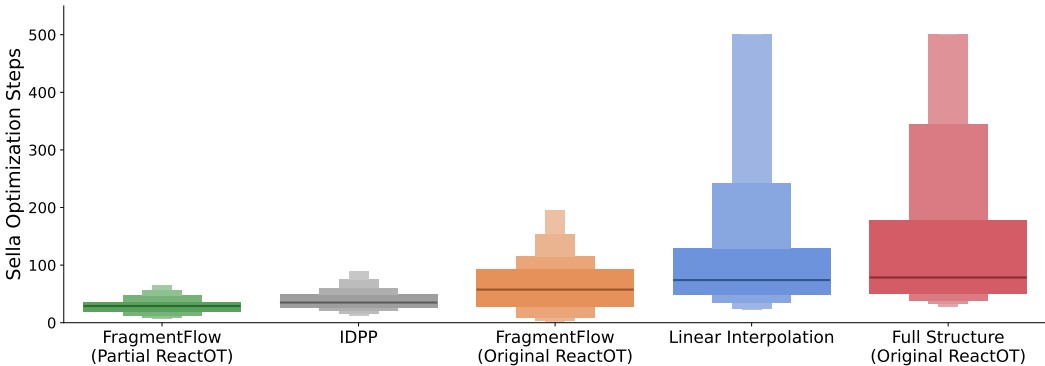

Figure 14: **Full Evaluation Results**. FragmentFlow (Partial ReactOT) yields the best TS initializations to the Sella optimizer, thus requiring the least number of Sella optimization steps among the tested methods.

It should be noted that the number of Sella optimization steps for FragmentFlow (Partial ReactOT) plateaus quite rapidly with respect to the number of NFEs used to generate the structure of the reactive core (see Figure 16 and Table 4). We hypothesize that this result stems from errors that arise from the attachment procedure, and leave this research direction for future work.

Table 3: **Comparison of TS Generation Methods.** Mean, median, and standard deviation of the number of Sella optimization steps required. Lower values indicate faster optimization.

| Method | Mean | Median | Std. Dev. |
|---|---|---|---|
| FragmentFlow (Partial ReactOT) | **30.51** | **29** | **18.58** |
| IDPP | 42.53 | 35 | 45.13 |
| FragmentFlow (Original ReactOT) | 70.11 | 57 | 67.52 |
| Linear Interpolation | 123.76 | 74 | 133.52 |
| Full Structure (Original ReactOT) | 145.91 | 78 | 141.37 |

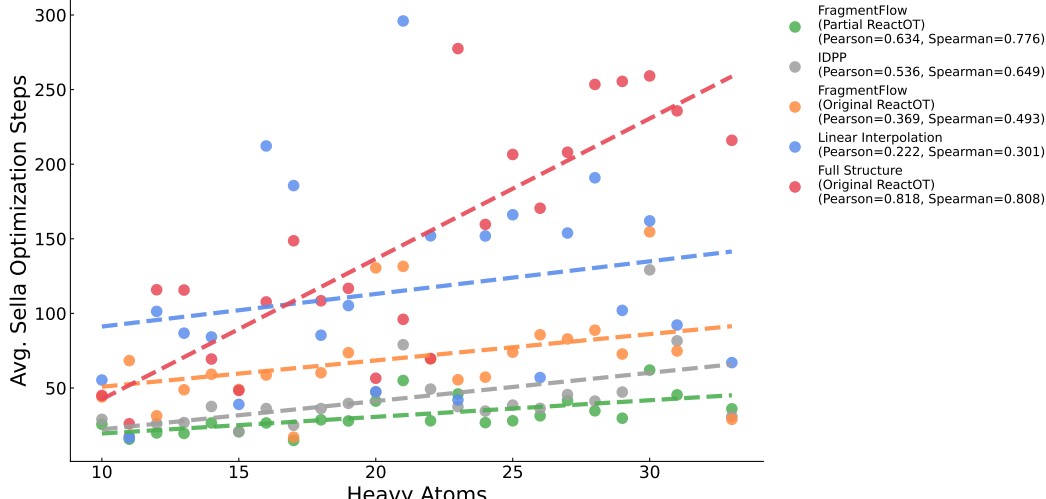

Figure 15: **Scaling Laws with Respect to the Number of Heavy Atoms.** Among all the tested methods FragmentFlow (Partial ReactOT) achieves the best performance. A notable trend is that Full Structure (Original ReactOT) gets increasingly worse with the size of the molecules (i.e., its performance diminishes as the distribution shift between the training set and the test set becomes more extreme).

Finally, we present complementary results to the model ablations in Section 4.1 that reinforce the conclusions drawn from the analysis in Section 4.3. Specifically, these results support the *core hypothesis*, which postulates that improvements in the quality of the generated reactive core reduce the number of Sella optimization steps required on average. Figure 17(a) shows that the RMSD to the reference reactive cores decreases as the number of random fragments included in the training

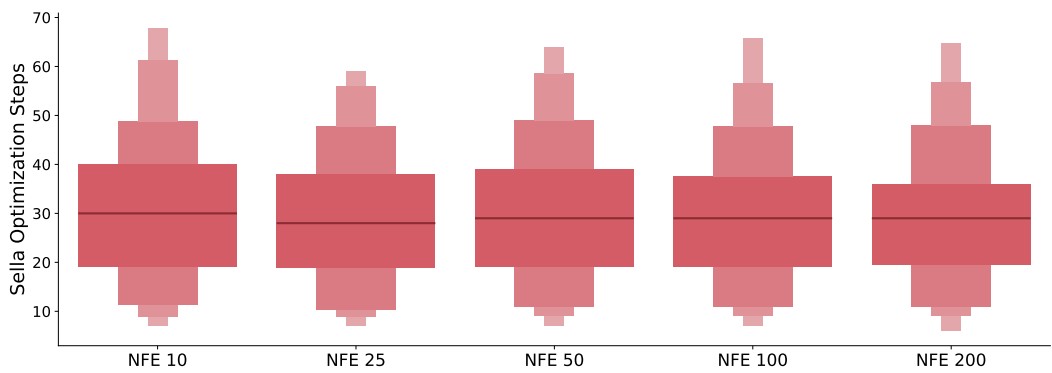

Figure 16: **The Effect of the NFEs Used to Generate the Reactive Core on the Quality of the Initial TS Guesses Produced by FragmentFlow (Partial ReactOT).** While an initial gain in performance is observed past 10 NFEs, the number of required Sella optimization steps plateaus after 25 NFEs.

Table 4: **Comparison of the NFEs Used to Generate the Reactive Cores.** Mean, median, and standard deviation of the number of Sella optimization steps required. Lower values indicate faster optimization.

| NFEs | Mean | Median | Std. Dev. |
|------|-------|--------|-----------|
| 10 | 31.10 | 30. | 17.10 |
| 25 | 30.18 | **28** | **16.51** |
| 50 | 30.60 | 29 | 17.25 |
| 100 | **30.00** | 29 | 16.69 |
| 200 | 30.51 | 29 | 18.58 |

set increases, while Figure 17(b) demonstrates a similar trend with respect to the number of training epochs. Finally, Figure 18 illustrates a strong correlation between the average RMSD of the reactive cores and the average number of Sella optimization steps required per data point. Together, these correlations provide further evidence for the *core hypothesis*, namely that more accurate reactive core geometries yield better TS initial guesses, as the reactive core exerts a dominant influence on the overall TS geometry.

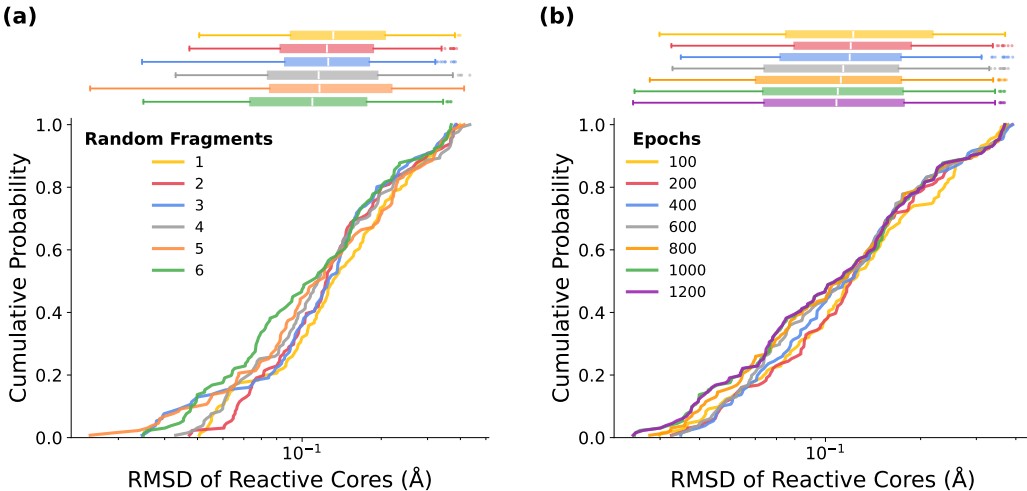

Figure 17: **Ablation Studies for Partial ReactOT: Examining the RMSD of the Reactive Cores. (a)** The reactive cores become more accurate with more random fragments in the training set, demonstrating the utility of this data augmentation approach. **(b)** The reactive cores become more accurate with more training epochs, but their quality plateaus after 1000 epochs.

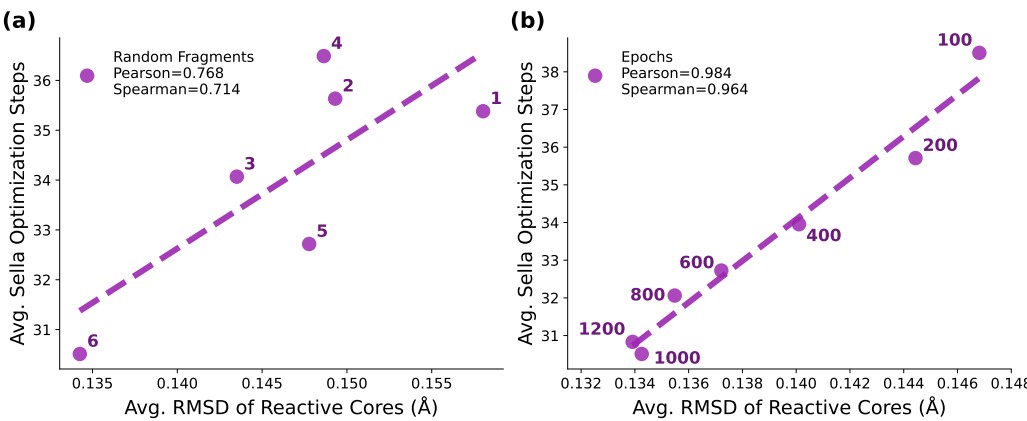

Figure 18: **Additional Evidence of the *Core Hypothesis*.** The average quality of the generated reactive core from Partial ReactOT correlates with the average number of required Sella optimization steps. This holds when examning the effect of **(a)** including more random fragments in the training set and **(b)** training Partial ReactOT for more epochs.

# E BENCHMARKING UMA

Machine learning interatomic potentials (MLIPs) provide quantum-mechanical accuracy at a fraction of the computational cost of DFT (Luise et al., 2025). MLIPs learn mappings from atomic configurations to potential energies and forces by training on databases of *ab initio* calculations. Recent advances have focused on developing "universal" MLIPs, such as UMA (Wood et al., 2025), capable of generalizing across diverse chemical spaces. In this work, we benchmark UMA for the TS generation problem and demonstrate that it can be successfully used in this setting while speeding up energy calculations by orders of magnitude without compensating accuracy. We use UMA in two core components of this work: first, to run energy evaluation during Sella optimization. Second, to evaluate the energies of our candidate structures. Overall, UMA enables us to evaluate energies in seconds, as compared to several hours that are required by traditional methods.

In this section, we benchmark UMA to justify its use. Specifically, we compare the $|\Delta E|$ values of the generated TSs relative to the reference structures as computed by UMA and by PySCF at the $\omega$B97X/6-31G$^*$ level of theory. As shown in Figure 19, the $|\Delta E|$ values obtained with UMA closely match those computed with PySCF, motivating our choice to use UMA in place of PySCF. Notably, this conclusion remains unchanged when Sella optimization is applied. In addition, we benchmark both the UMA-s and UMA-m model sizes and show in Figure 20 that they yield very similar results on the validation dataset of Transition1x. Therefore, we adopt UMA-s, as it offers comparable accuracy while being smaller and faster.

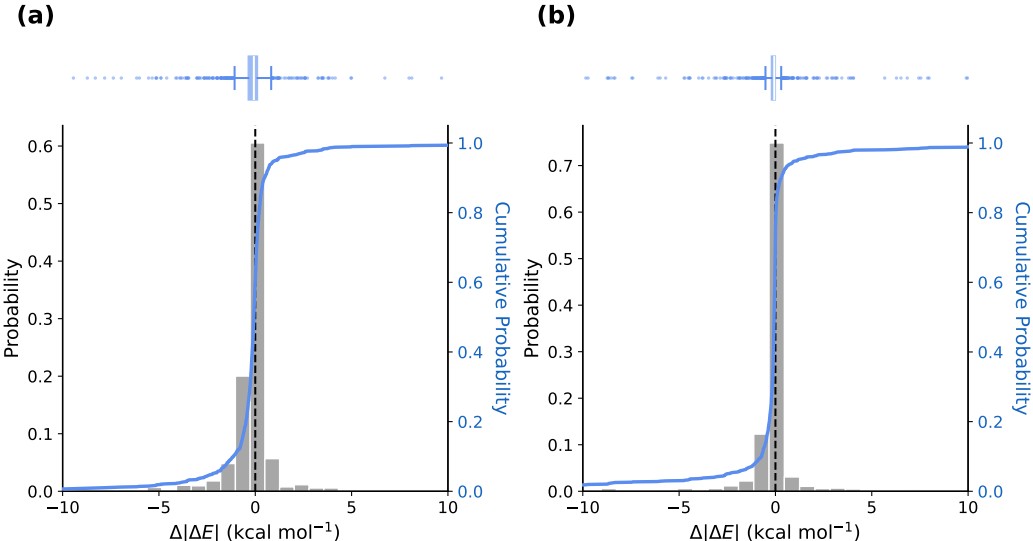

Figure 19: **Benchmarking UMA.** Energies were evaluated on the Transition1x validation set using Original ReactOT (NFE = 200) to generate candidate samples and using PySCF and UMA-s for energy evaluations. The distribution of the differences between PySCF and UMA-s calculated $|\Delta E|$ is presented **(a)** before and **(b)** after running Sella optimization on the generated samples.

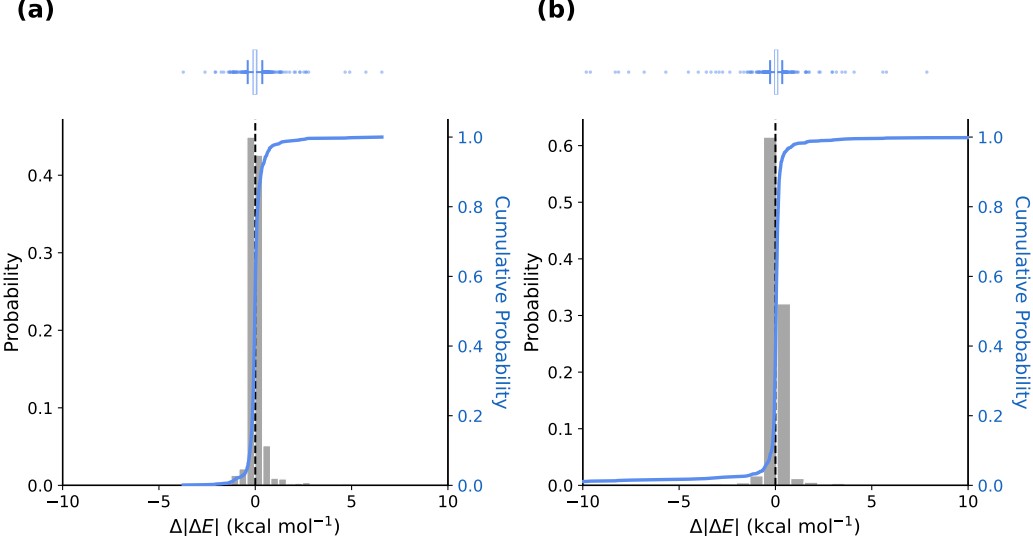

Figure 20: **UMA Model Size.** Energies were evaluated on the Transition1x validation set using Original ReactOT (NFE = 200) to generate samples and using UMA-s and UMA-s for energy evaluations. The distribution of the differences between UMAS-s and UMA-m calculated $|\Delta E|$ is presented **(a)** before and **(b)** after running Sella optimization on the generated samples.

## F    EXAMPLE STRUCTURES

We provide qualitative evaluations by presenting sample structures that were used during central steps of our analysis. Figures 21 and 22 demonstrate the results of our automatic reactive core identification procedure as discussed in Section 3.4 and Appendix C. Figure 23 shows the structures of the substituents that we used for the curation of LargeT1x. Laslty, Figure 24 shows sample reactive cores and full structures (both before and after Sella optimization) that were generated using IDPP and FragmentFlow (Partial ReactOT).

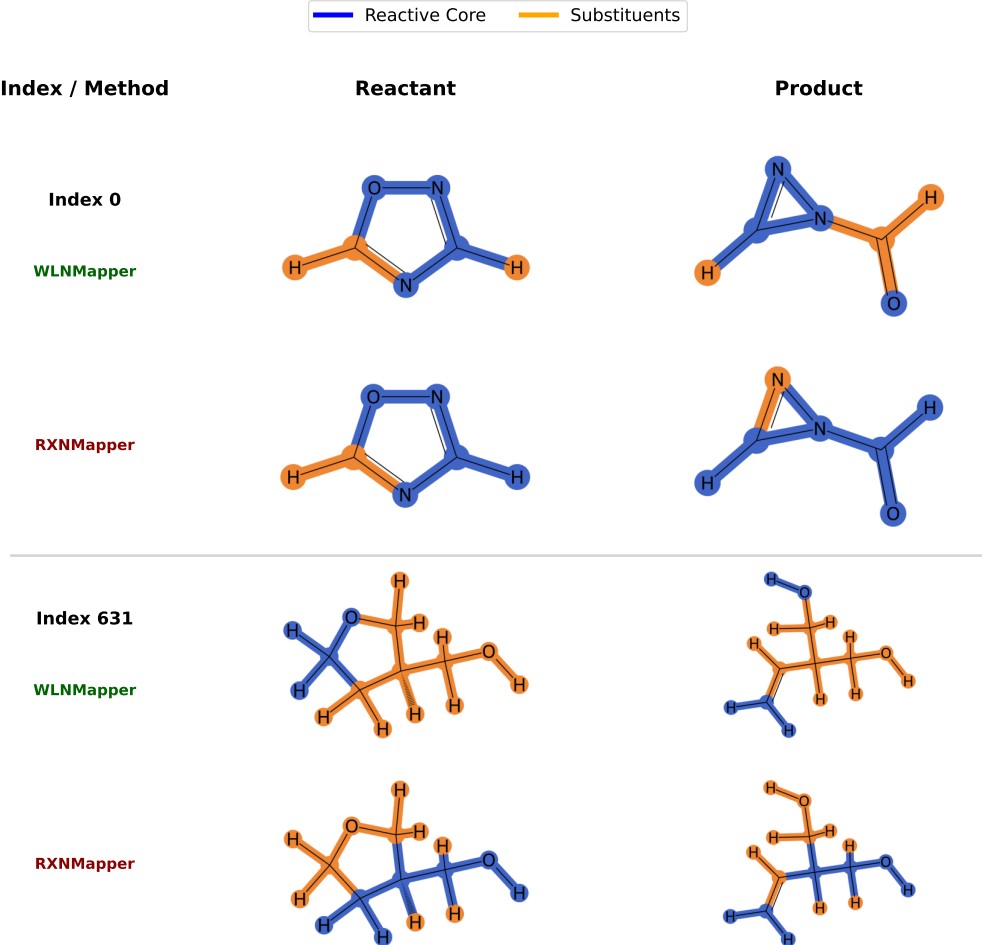

Figure 21: **Comparing Atom Mappers.** WLN mapper is generally more successful at identifying the reactive core.

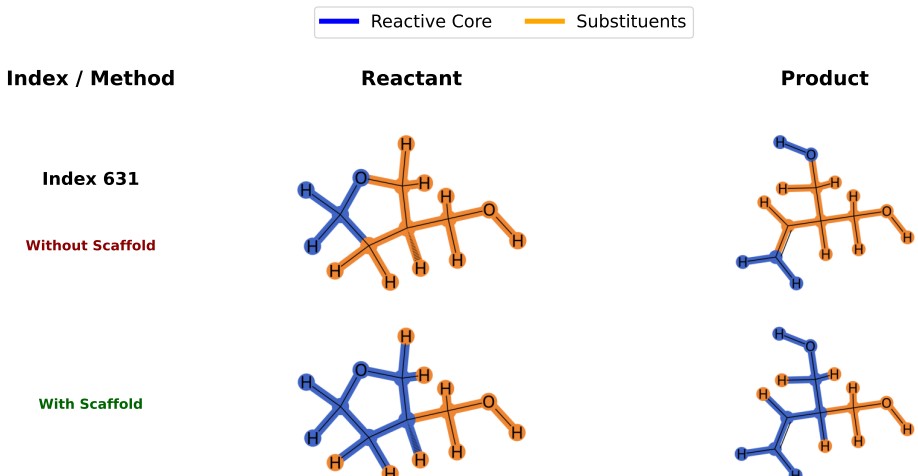

**Reactive Core Identification: Without Scaffold vs With Scaffold**

Figure 22: **Backbone Scaffold.** Taking the union between the reactive core as identified by WLN mapper and the atoms from the Bemis–Murcko scaffold empirically increases the quality of our reactive core identification procedure. We consider our final reactive core to be this union.

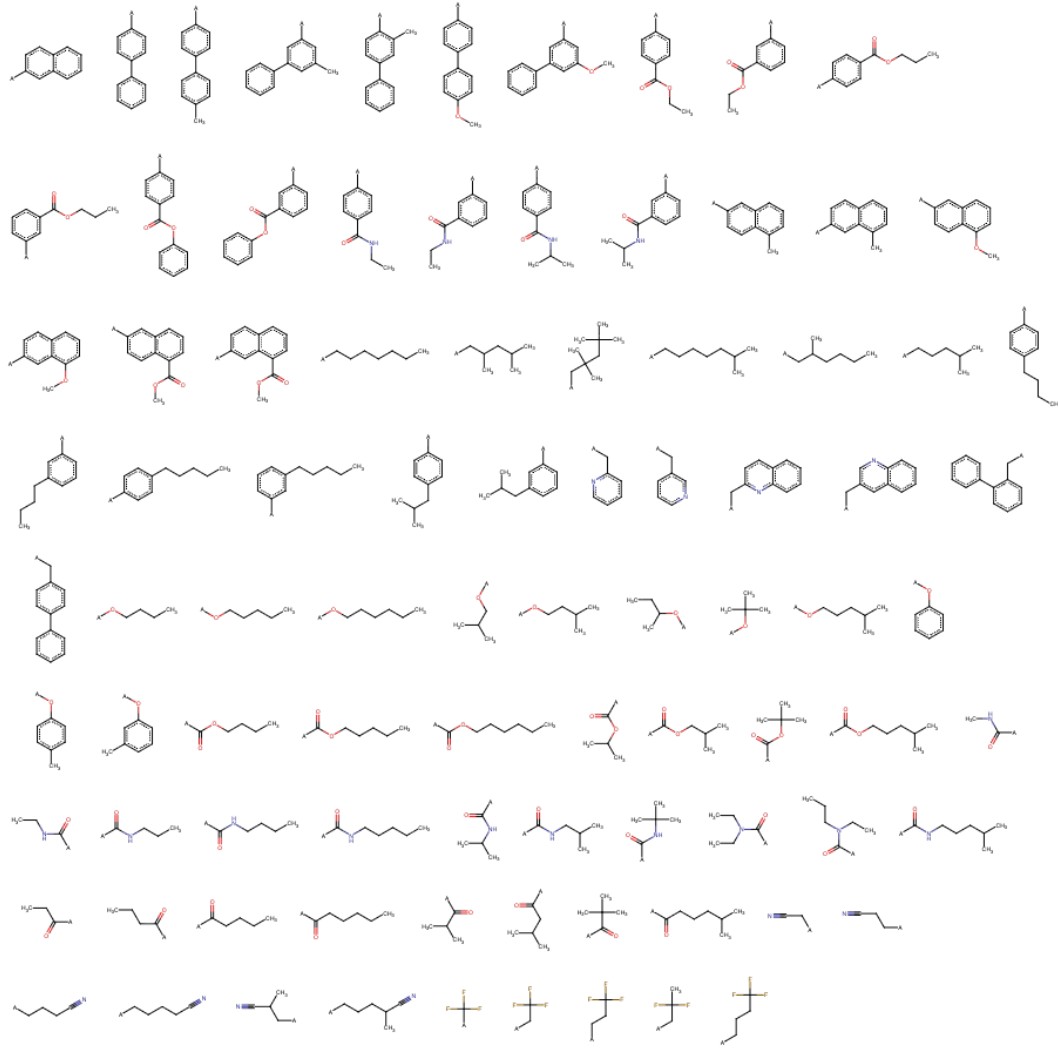

Figure 23: **Substituent Structures.** The set of substituents that was used for the curation of LargeT1x. The attachment point to the reactive core is denoted with an asterisk.

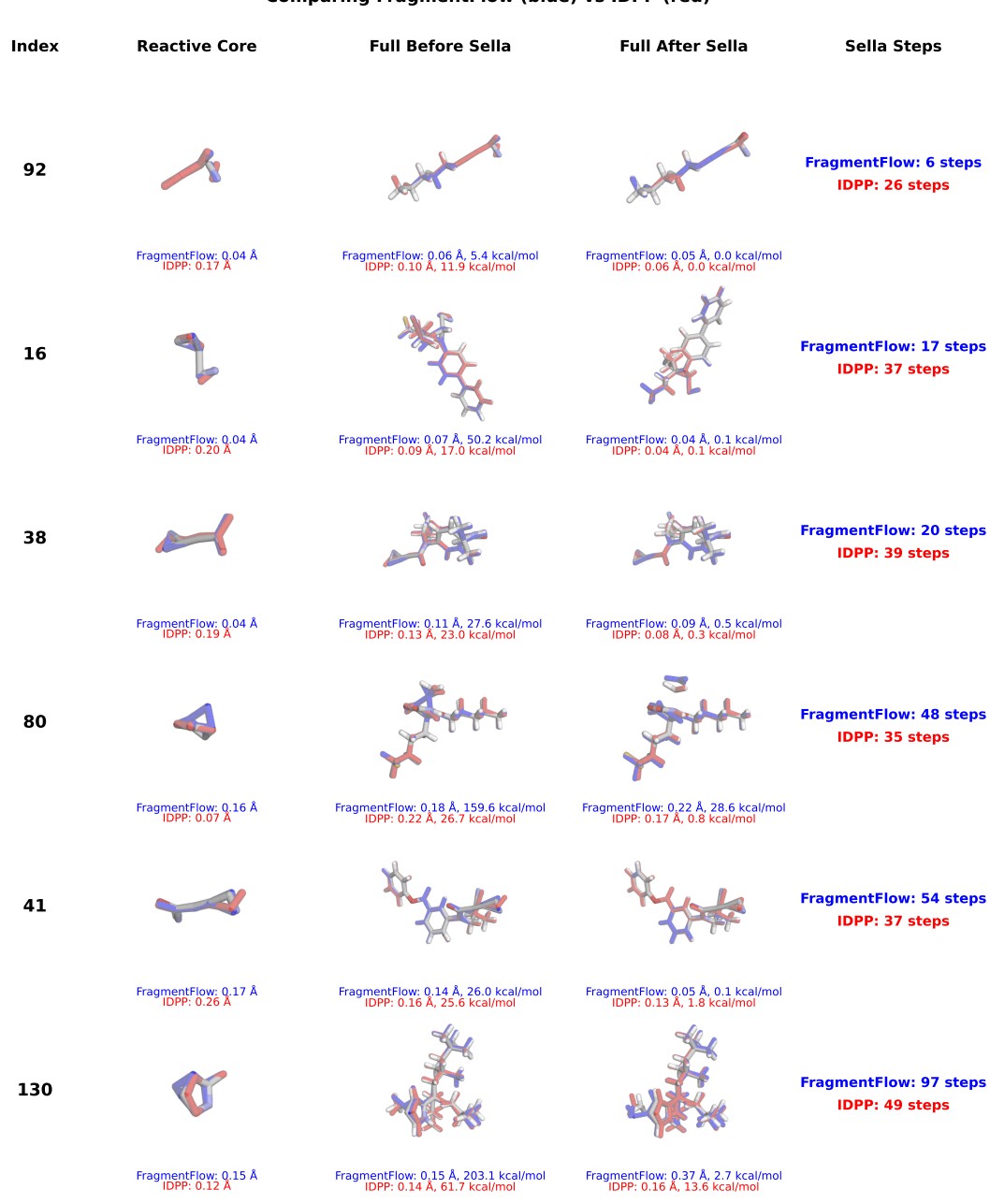

Figure 24: **Example Structures**. Reactive core and full structures (before and after Sella) for IDPP red and FragmentFlow (Partial ReactOT). Reference structures are shown using default atom type coloring. The first three structures (indices 92, 16, and 38) show cases where FragmentFlow requires less Sella optimization steps than IDPP. The last three structures (indices 80, 41, and 130) show cases where IDPP requires less Sella optimization steps than FragmentFlow.

