# OpenReview forum: "FragmentFlow: Scalable Transition State Generation for Large Molecules"
_ICLR.cc/2026/Workshop/FM4Science — ICLR 2026 Workshop FM4Science Poster_

### Official Review · Reviewer_ef4Q · 2026-02-19
**Strong Contribution to Scalable Transition State Generation Aligned with FM4Science Themes**

**Rating:** 8
**Confidence:** 4

**Review:**

The paper proposes FragmentFlow, a divide-and-conquer approach for generating transition states (TSs) in chemical reactions involving large molecules. It addresses distribution shifts in generative models (specifically flow matching based on ReactOT) caused by increasing molecular size by focusing generation on the reactive core and re-attaching substituents via classical methods like IDPP interpolation, followed by refinement with the Sella optimizer. The authors curate a new benchmark dataset, LargeT1x, with 131 reactions involving up to 33 heavy atoms. Evaluations show that FragmentFlow achieves ~90% of TSs within 1 kcal/mol of reference energies after optimization, with 30% fewer Sella steps than classical baselines like IDPP, and better scaling with molecular size. The work validates the hypothesis that reactive core quality drives overall TS accuracy.
Strengths

Relevance to Workshop Themes: This work aligns well with FM4Science's focus on foundation models tailored for science. It adapts flow matching (a generative paradigm) for chemistry, incorporating scientific priors (locality of reactive sites) to improve out-of-distribution generalization to larger molecules—a key challenge in scaling laws for scientific data. It also demonstrates integration with classical tools (IDPP, Sella, UMA), bridging ML with scientific workflows, and emphasizes evaluations on real-world metrics like optimization efficiency and energy accuracy.
Novel Contributions: The fragmentation strategy is a clever way to mitigate size-induced distribution shifts without requiring massive new TS data for large molecules. The new LargeT1x dataset extends existing benchmarks (e.g., Transition1x) to practically relevant scales, enabling empirical validation. Ablations on data augmentation (random fragments) and correlations between core quality and full TS performance provide mechanistic insights.
Technical Soundness and Rigor: The method is well-motivated theoretically (KL divergence analysis for factorized modeling) and empirically validated. Training recipes are reproducible, with hyperparameters detailed. Evaluations use meaningful metrics (RMSD, |ΔE|, Sella steps) and compare against strong baselines (ReactOT, IDPP, linear interpolation). Scaling analysis with molecular size is particularly insightful for scientific applications.
Potential Impact: By reducing optimization costs, this could enable high-throughput reactivity studies in drug discovery or materials design, accelerating scientific discovery as per the workshop's aims.

Limited Scope and Scale: The evaluation dataset is small (131 reactions from 33 unique ones), potentially limiting generalizability across diverse reaction types. While the strict validation (IRC + frequency analysis) ensures quality, it might introduce selection bias. No discussion of failure modes or broader chemical spaces (e.g., beyond organic reactions).
Lack of Uncertainty Quantification: The model is deterministic (fixed initial condition in ReactOT), and there's no explicit handling of uncertainty in TS predictions, which is a priority in the workshop's principles for reliable scientific models. Future work mentions ensembles, but this is not explored here.
Dependence on Classical Methods: While integration with tools like Sella is a strength, the approach still relies heavily on expensive post-generation optimization, which might not fully realize "foundation model" efficiency for end-to-end use. Comparisons to other generative baselines (e.g., diffusion models) are limited.
Minor Presentation Issues: Some citations appear futuristic (e.g., 2025/2026 dates), possibly placeholders. The paper could better connect to broader FM principles, like multi-modality or causal structure in reactions.

How sensitive is performance to reactive core identification? What if automatic extraction fails more often in diverse chemistries?
Could you quantify wall-clock time savings more precisely (e.g., on GPU/CPU setups) to highlight high-throughput potential?
Plans to release LargeT1x and code? This would aid reproducibility and community benchmarking.
How does FragmentFlow handle multi-step reactions or ensembles of TSs, as hinted in future work?

This is a strong submission that advances generative modeling for chemical sciences, directly addressing workshop aims like scaling, OOD generalization, and hybrid ML-classical systems. It provides principled innovations and empirical evidence, making it suitable for presentation to foster discussion on scientific FMs. Minor expansions on UQ and broader evaluations would strengthen it further.

---

### Official Review · Reviewer_BbbK · 2026-02-22

**Rating:** 7
**Confidence:** 4

**Review:**

**Summary:**

This paper studies transition state (TS) generation for larger molecules. The authors observe that existing TS generators degrade as molecule size increases, partly due to distribution shift and limited large-molecule TS data. To this end, they propose FragmentFlow, generating TS geometry of the reactive core while keeping substituents fixed. The method is evaluated on extended TS benchmarks and compared against linear interpolation, IDPP, and pretrained ReactOT models. Performance is measured by both RMSD and downstream saddle-point optimization using Sella and IRC validation.

**Strengths:**

1. Instead of generating full TS structures for large molecules, the model predicts only the reactive core geometry and then reattaches substituents. This reduces distribution shift and makes the learning problem more controlled.

2. In addition to geometric error, the authors run Sella optimization and verify IRC connectivity. This makes the reported success rate more meaningful.

3. A task-specific dataset with large molecules is curated, extending existing TS benchmarks. The authors also provide a detailed dataset construction procedure.

4. The paper's visual design is aesthetically pleasing.

**Weaknesses and questions:**

1. FragmentFlow (Partial ReactOT) is clearly better than Full Structure (Original ReactOT), which is consistent with reduced distribution shift on larger molecules. But the improvement relative to IDPP looks marginal. Did authors conduct statistical testing to show the improvement is statistically significant?

2. Several plots include error bars, and Table 3-4 show mean and standard deviation, but the paper does not state how repetitive experiments were conducted. Please clarify the repetition protocol (number of runs, random seeds).

3. Can the authors disentangle how much of the gain comes from the reactive-core restriction itself versus the specific learning objective and training procedure of the flow model? Clarifying this would help better understand whether the improvement mainly stems from the problem decomposition or from the particular modeling strategy.

4. The paper decomposes the KL divergence into a reactive-core term and an attachment term, and hypothesizes that the attachment error grows slowly with molecular size. Beyond the qualitative discussion, can the authors quantify how the reactive core modeling error and the attachment error (or suitable proxies) scale with reactive-core size $N_c$ and overall molecular size $N$?

5. The caption of Figure 5 mentions Pearson and Spearman correlations, but the variables involved are not explicitly stated.

6. Minor writing issues remain (e.g., in line 458, “less Sella optimization steps” should be “fewer”), and the full name of RMSD is not clarified at first mention.

---

### Decision · Program_Chairs · 2026-03-03

Accept (Poster)